# Environmental deformations dynamically shift the grid cell spatial metric

Alexandra T Keinath[1]*, Russell A Epstein[1], Vijay Balasubramanian[2]*

[1]Department of Psychology, University of Pennsylvania, Pennsylvania, United States; [2]Department of Physics, University of Pennsylvania, Pennsylvania, United States

**Abstract** In familiar environments, the firing fields of entorhinal grid cells form regular triangular lattices. However, when the geometric shape of the environment is deformed, these time-averaged grid patterns are distorted in a grid scale-dependent and local manner. We hypothesized that this distortion in part reflects dynamic anchoring of the grid code to displaced boundaries, possibly through border cell-grid cell interactions. To test this hypothesis, we first reanalyzed two existing rodent grid rescaling datasets to identify previously unrecognized boundary-tethered shifts in grid phase that contribute to the appearance of rescaling. We then demonstrated in a computational model that boundary-tethered phase shifts, as well as scale-dependent and local distortions of the time-averaged grid pattern, could emerge from border-grid interactions without altering inherent grid scale. Together, these results demonstrate that environmental deformations induce history-dependent shifts in grid phase, and implicate border-grid interactions as a potential mechanism underlying these dynamics.
DOI: https://doi.org/10.7554/eLife.38169.001

## Introduction

The hippocampal formation is thought to maintain a metric representation of space that preserves distances between represented locations, sometimes referred to as a cognitive map (*O'Keefe and Nadel, 1978*; *O'Keefe and Dostrovsky, 1971*). Entorhinal grid cells are hypothesized to generate this metric via an internally updated, path-integrated representation of space (*Hafting et al., 2005*; *McNaughton et al., 2006*; *Moser and Moser, 2008*; *Buzsáki and Moser, 2013*; *Moser et al., 2014*; *Fuhs and Touretzky, 2006*). Results of environmental deformation experiments have led to the belief that this metric is fundamentally malleable (*Barry et al., 2007*; *Krupic et al., 2016*; *Stensola et al., 2012*; *Krupic et al., 2018*). In these experiments, neural activity is recorded as a rat explores a familiar environment that has been modified by stretching, compressing, or removing/inserting chamber walls. Such deformations induce a number of distortions in the time-averaged activity of both grid cells (*Barry et al., 2007*; *Stensola et al., 2012*) and hippocampal place cells (*O'Keefe and Burgess, 1996*; *Gothard et al., 1996*; *Barry et al., 2006*; *Muller and Kubie, 1987*; *Lever et al., 2002*). Often described as 'rescaling', these distortions have been taken to suggest that the spatial metric of the cognitive map can be reshaped by altering environmental geometry (*Barry et al., 2007*; *Sheynikhovich et al., 2009*; *Raudies et al., 2016*). This interpretation assumes that the distortions observed in the time-averaged rate maps of grid and place cells reflect fixed changes to the underlying spatial code that are independent of the movement history of the navigator. Here, we present results that challenge this assumption, and indicate that the grid cell spatial metric undergoes dynamic history-dependent phase shifts during environmental deformations.

In a familiar environment, local navigational boundaries play an important role in anchoring the grid code. For example, the grid representation is most precise when the navigator has recently encountered a boundary (*Hardcastle et al., 2015*). When the environment is moved relative to the broader reference frame of the experimental room, grid phase often remains anchored to the local

*For correspondence:
atkeinath@gmail.com (ATK);
vijay@physics.upenn.edu (VB)

**Competing interests:** The authors declare that no competing interests exist.

boundaries (*Savelli et al., 2017*). At a circuit level, these dynamics may be driven by input to grid cells from border cells. Co-localized with grid cells in the entorhinal cortex, border cells are active only when a boundary is nearby and at a particular allocentric direction. Simulations have demonstrated that border input can be sufficient to reset grid phase and stabilize the grid pattern, provided that border inputs coding for different boundaries map to a consistent allocentric grid phase (*Hardcastle et al., 2015*; *Pollock et al., 2018*).

During an environmental deformation, the familiar arrangement of boundaries is altered. In turn, the relationship among the firing fields of border cells is also altered: stretching or compressing a boundary yields a concomitant rescaling of border activity, displacing a boundary also displaces neighboring border fields, and inserting a new boundary elicits additional border fields at analogous locations neighboring both boundaries (*Solstad et al., 2008*; *Savelli et al., 2008*). This entails that a mapping of border input to grid phase adapted to the familiar environment will no longer signal a consistent allocentric grid phase. This predicts that during a deformation grid phase will dynamically shift as the navigator encounters different boundaries. Importantly, such boundary-tethered shifts in grid phase would be obscured during traditional time-averaged analyses in which data following encounters with different boundaries are pooled together. However, averaging over such shifts would also contribute to distortions of the time-averaged pattern. Thus, grid pattern distortions during environmental deformations may be in part driven by input from border cells (*Krupic et al., 2016*; *Hardcastle et al., 2015*; *Solstad et al., 2008*; *Giocomo, 2016*; *Bush et al., 2014*; *Cheung, 2014*), specifically through dynamic history-dependent shifts in grid phase throughout exploration.

To test this prediction, we reexamined datasets from two previous environmental deformation experiments (*Barry et al., 2007*; *Stensola et al., 2012*). Our analyses revealed direct evidence of boundary-tethered shifts in grid phase during environmental deformations. These shifts were previously unrecognized and contribute to the appearance of rescaling in the time-averaged rate maps. Next, we implemented a computational model of border cell-grid cell interactions which reproduced these dynamics. These simulations further demonstrated that boundary-tethered shifts in grid phase can interact with the particular path of the navigator to give rise to grid scale-dependent rescaling and local distortions of the time-averaged grid pattern, as observed experimentally (*Barry et al., 2007*; *Stensola et al., 2012*; *Krupic et al., 2018*). Together, these results demonstrate that geometric deformations of a familiar environment induce history-dependent shifts in grid phase and implicate border cell-grid cell interactions as a potential source of these dynamics.

## Results

### Boundary-tethered shifts in grid phase are observed in recorded grid cells during rescaling deformations

To test whether boundary-tethered shifts in grid phase are observed during environmental deformations, we reanalyzed data from two classic deformation studies ((*Barry et al., 2007*) and (*Stensola et al., 2012*)). In *Barry et al., 2007*), rats were familiarized with either a 100 cm x 100 cm square or a 100 cm x 70 cm rectangular open environment, and then reintroduced to deformed and undeformed versions of these environments (i.e. all combinations of chamber lengths and widths of 70 cm or 100 cm), while the activity of grid cells was recorded (familiar square: 23 grid cells; familiar rectangle: 13 grid cells meeting criteria; see Materials and methods). In *Stensola et al. (2012)*, rats were familiarized with a 150 cm x 150 cm square open environment, and then reintroduced to deformed (100 cm x 150 cm rectangular) and undeformed versions of this environment, while data were recorded from 51 grid cells. As previously reported (*Barry et al., 2007*; *Stensola et al., 2012*), the time-averaged grid patterns of most of these grid cells appeared to rescale when the environment was deformed, though the grid patterns of a subset of cells with smaller grid scales did not.

First, we separated the spiking data of each cell according to the most recently contacted boundary, either the north, south, east or west, with contact defined as coming within 12 cm of that boundary (*Hardcastle et al., 2015*). From these data, we created four boundary rate maps which summarized the spatial firing pattern of the grid cell after contacting each boundary. Comparison of such rate maps, conditioned on contact with opposing boundaries (north-south vs. east-west), revealed clear examples of grid shift along deformed dimensions (*Figure 1*). To quantify shift

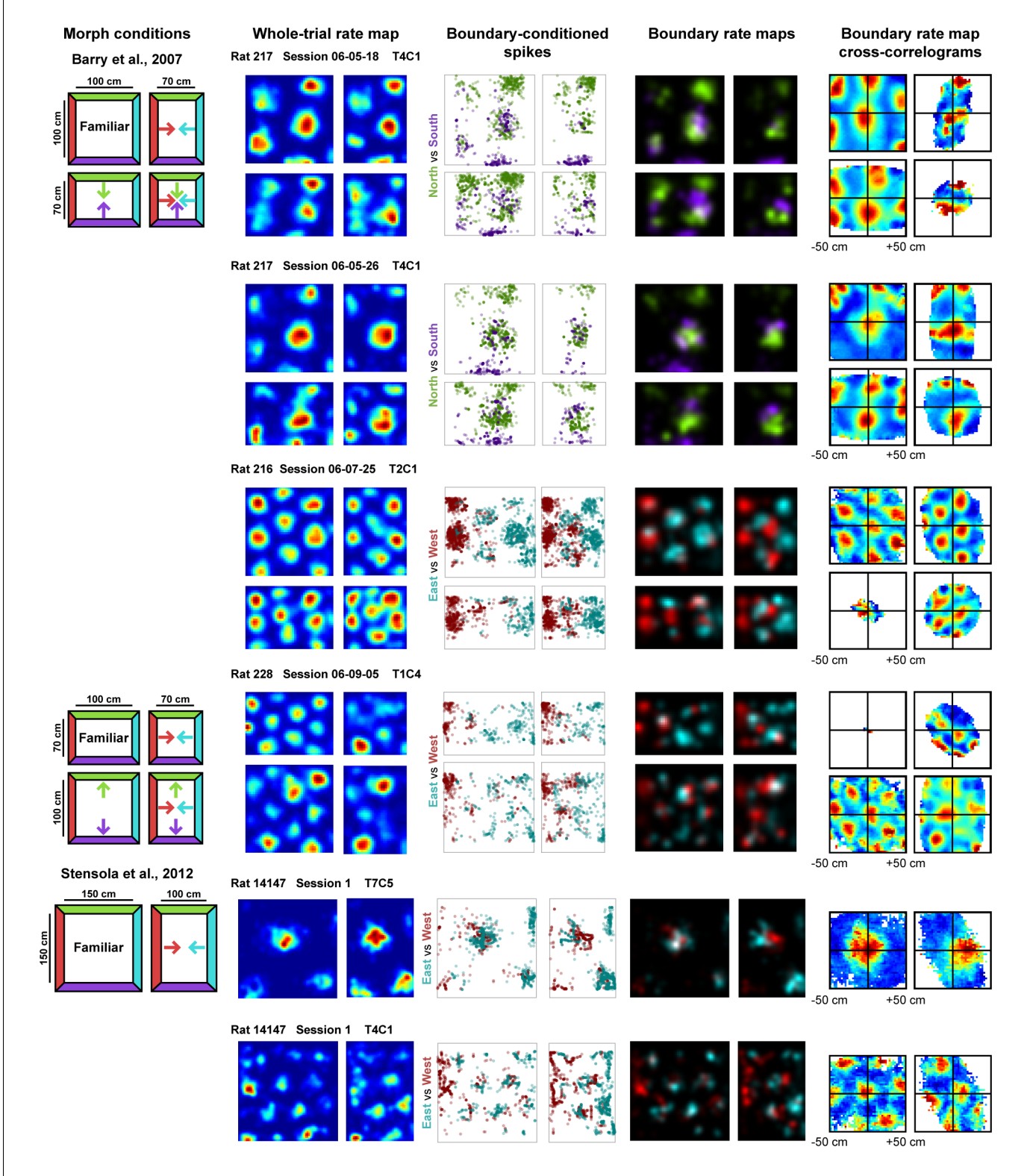

**Figure 1.** Examples of whole trial rate maps, boundary-conditioned spikes, boundary rate maps, and cross-correlograms of opposing boundary rate maps for recorded grid cells. Rat, session, and cell identity indicated above whole trial rate maps. Boundary-conditioned spikes and boundary rate maps organized by opposing north-south (green—purple) and east-west (blue—red) boundary pairs. Colored arrows in morph condition indicate the shifts predicted by the boundary-tethered model during each deformation.

DOI: https://doi.org/10.7554/eLife.38169.002

separately for each dimension, we cross-correlated the opposing boundary rate map pairs (i.e. north-south or east-west boundary pairs) and computed the distance from center of the cross-correlogram (0,0 lag) to the peak nearest the center along the dimension of interest (see Materials and methods). To ensure that any shifts observed were not the product of differential spatial sampling following each boundary contact, we randomly selected subsets of the data to match the number of samples at each pixel prior to constructing each opposing boundary rate map for this analysis (*Figure 2—figure supplement 1*). The mean shift across 100 iterations of this analysis was then taken as the final measurement of shift along that dimension.

Even in a familiar environment, finite sampling noise will cause this measure of shift to be non-zero. Compared to this baseline, grid shift increased along deformed, but not undeformed, dimensions (*Figure 2A*; *Figure 2—source data 1*). An increased shift was observed even in cells with small-scale grid patterns (*Figure 2B*), as well as cells whose time-averaged grid patterns did not appear to rescale (*Figure 2C*). This increase in shift along deformed dimensions was not due to differences in sampling during deformation trials (*Figure 2—figure supplement 1*). In sum, these results indicate that deformation-induced phase shifts are observed in grid cells regardless of their scale or the extent to which their time-averaged rate maps resemble a rescaling.

Next, we tested whether the grid pattern of each boundary rate map maintained its familiar phase relative to the corresponding boundary. To address this question, we compared each boundary rate map to the whole-trial familiar environment rate map, while varying the alignment of the two maps along the deformed dimension. If the spatial relationship relative to the most recently contacted boundary is preserved, then each boundary rate map should be most similar to the familiar

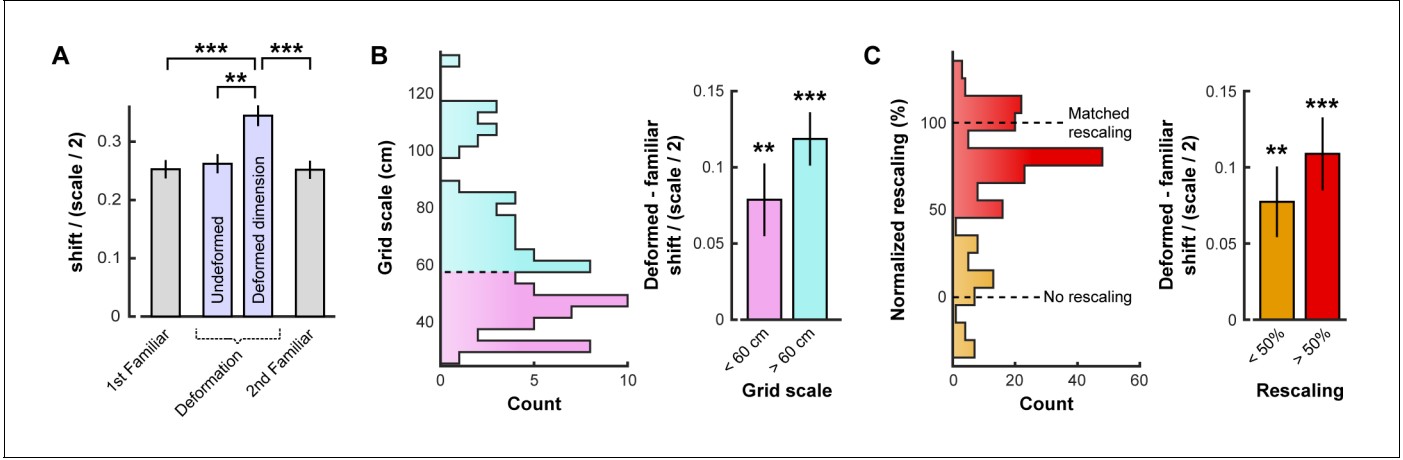

**Figure 2.** Opposing boundary rate maps are relatively shifted in phase along deformed dimensions during deformations. Data from all experiments in (*Barry et al., 2007*; *Stensola et al., 2012*) combined. All error bars denote mean ±SEM. (**A**) Grid shift as measured by the relative phase between opposing boundary rate maps along deformed and undeformed dimensions. Because of the periodic nature of the grid pattern, shift is bounded by one half of the grid scale. Therefore, we quantify shift as a ratio of shift / (scale/2). (paired t-tests, 1st familiar vs. deformed: t(80) = 5.44, p<0.001; undeformed vs. deformed: t(83) = 4.21, p=0.001; 2nd familiar vs. deformed: t(82) = 5.07, p<0.001; all other comparisons: t < 1.10, p>0.2731; *Figure 2—source data 1*). (**B**) Histogram of grid scale (left) and shift ratio separated by grid scale along deformed dimensions with mean familiar shift ratio subtracted (right). (t-test versus 0, scale <60 cm: t(45) = 3.43, p=0.001; scale >60 cm: t(40) = 4.67, p<0.001; *Figure 2—source data 1*). (**C**) Histogram of rescaling (left) and shift ratio separated by rescale along deformed dimensions with mean familiar shift ratio subtracted (right). Because rescaling could vary within cell along deformed dimensions, each condition and deformed dimension was treated independently. (t-test versus 0, rescaling <50%: t(61) = 3.39, p=0.001; rescaling >50%: t(130) = 7.05, p<0.001; *Figure 2—source data 1*). **p<0.01, ***p<0.001.
DOI: https://doi.org/10.7554/eLife.38169.003

The following source data and figure supplement are available for figure 2:

**Source data 1.** Results of grid shift analyses.
DOI: https://doi.org/10.7554/eLife.38169.005
**Source data 2.** Results of phase alignment analysis.
DOI: https://doi.org/10.7554/eLife.38169.006
**Figure supplement 1.** Controlling for sampling biases when measuring boundary-tethered shift.
DOI: https://doi.org/10.7554/eLife.38169.004

environment rate map when the two maps are aligned by the corresponding boundary. If, on the other hand, reshaping a familiar environment rescales the grid pattern symmetrically, then the familiar and boundary rate maps should be equally well aligned by either the corresponding or the opposite boundary. Consistent with the boundary-tethered prediction, we found that the correlation between the deformed environment boundary rate map and the familiar environment rate map was maximized when the two maps were aligned by the corresponding boundary rather than the opposite boundary (313 of 390 comparisons; sign test versus 50%: p<0.001; *Figure 2—source data 2*).

Boundary-tethered shifts in grid phase may contribute to the appearance of rescaling. If so, then the appearance of rescaling should be reduced when the data are divided according to the most recently contacted boundary. In contrast, if boundary-tethered shifts do not contribute to the appearance of rescaling, then a similar amount of rescaling should be observed regardless of whether or not data are conditioned on the most recently contacted boundary. To test these predictions, we computed the grid rescaling factor between the familiar rate map and each deformed-dimension boundary rate map, aligned by the corresponding boundary. For comparison, we also computed the grid rescaling factor between the familiar rate map and the whole-trial rate map, aligned by the same boundary. To ensure that any reduction in rescaling observed was not the product of differential spatial sampling following boundary contact, we randomly selected subsets of the whole-trial data to match the sampling distributions of the whole-trial rate map and boundary rate map for each comparison (*Figure 2—figure supplement 1*). The mean rescaling across 100 iterations of this analysis was then taken as the final measurement of whole-trial rate map rescaling. This analysis revealed a significant reduction in the appearance of rescaling following boundary-conditioning (*Figure 3A*; *Figure 3—source data 1*), consistent with a contribution of boundary-tethered phase shifts to the appearance of rescaling in grid cells.

We next tested whether environmental deformations affect grid field size. Boundary-tethered phase shifts, when averaged over the whole trial, should yield an increase in field length primarily along deformed dimensions, regardless of whether the environment was compressed or stretched. On the other hand, a pure rescaling of the grid pattern predicts an increase in field length during stretching deformations, but a decrease in field length during compressions. Because both accounts predict an increase in field length during stretching deformations, we focused on compression trials. From the whole-trial rate maps of each cell, we computed the field length during compression deformations, separately along deformed and undeformed dimensions. This analysis revealed an increase in field length along deformed, but not undeformed, dimensions relative to field length in the familiar environment (*Figure 3B*; *Figure 3—source data 2*), consistent with the boundary-tethered prediction. For completeness, we also examined stretching deformations. Field length along deformed dimensions also increased numerically during these deformations (mean ±SEM, familiar: 33.27 ± 5.39 cm; deformed: 34.81 ± 4.17 cm), although this effect did not reach significance in this small sample (n = 13; paired t-test: t(12) = 0.22, p=0.828).

We then examined whether deformations affected firing rates in a manner consistent with the boundary-tethered phase shift prediction. If, during deformations, grid vertices are shifted to different locations when different boundaries are encountered, then averaging across trajectories originating from multiple boundaries will necessarily reduce the peak values of the whole trial rate map. Thus, the boundary-tethered model predicts a reduction in the peak firing rate during environmental deformations, as measured by the maximum value of the whole-trial rate map. On the other hand, because the density of grid fields within the environment remains unchanged on average, grid shift does not predict a change in mean firing rate, as measured by the total number of spikes across the entire trial divided by the trial duration. Although a pure rescaling account does not make specific predictions about peak and mean firing rates, the simplest assumption would be that neither should change, as the density and intensity of fields tiling the space should be preserved during deformations (*Ismakov et al., 2017*). Consistent with the boundary-tethered phase shift prediction, peak firing rates were significantly reduced during deformation trials relative to familiar trials while mean firing rates did not differ significantly (*Figure 3C*; *Figure 3—source data 3*).

Finally, we tested whether deformed rate maps could be accurately predicted by boundary-tethered phase shifts on a trial-by-trial basis. To do so, for each cell and deformation trial we first created predicted boundary rate maps for each displaced boundary from the familiar environment rate map. These rate maps were shifted versions of the familiar rate map, aligned by the corresponding boundary (*Figure 3—figure supplement 1*). If the length of a boundary changed, then the central

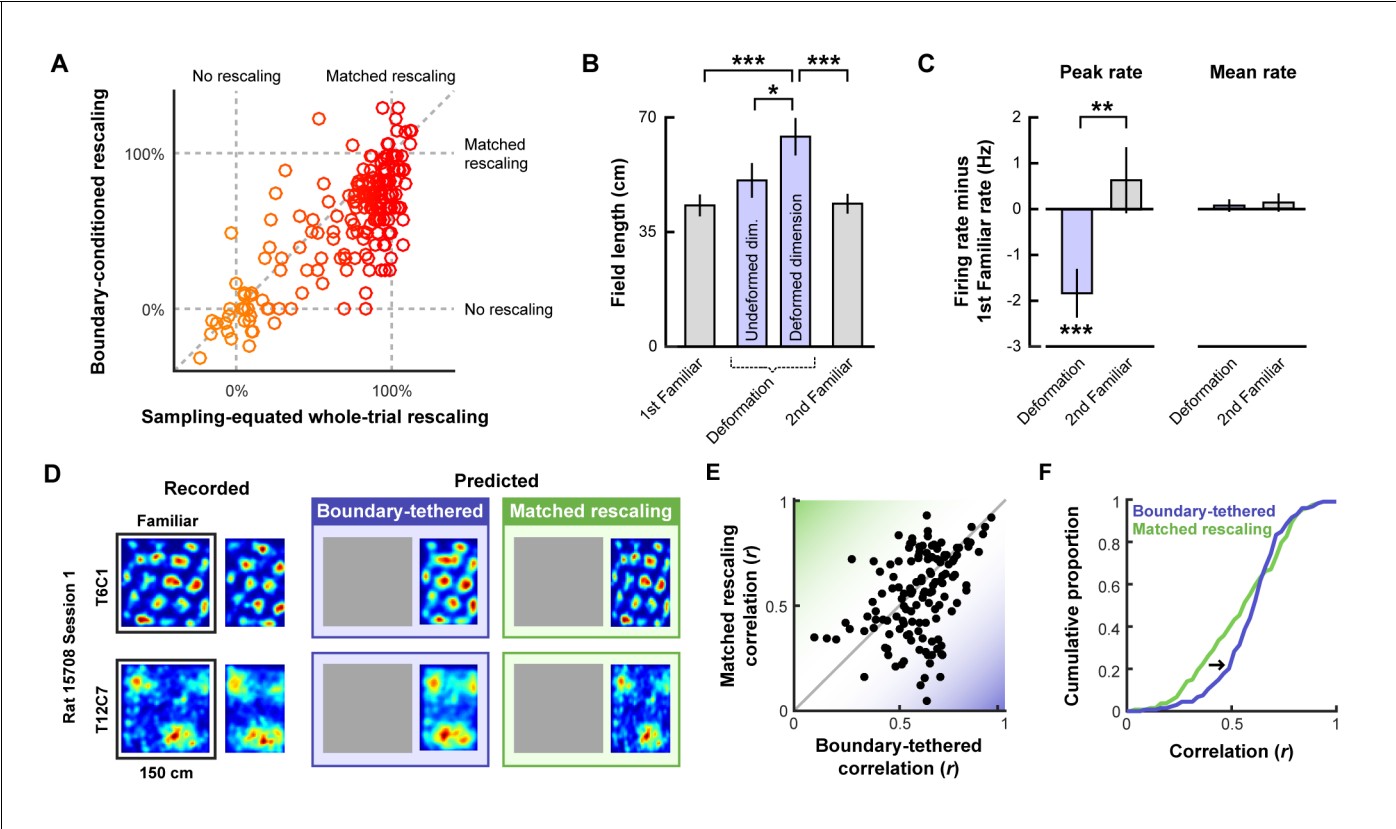

**Figure 3.** Additional tests of boundary-tethered phase shift predictions. Data from all experiments in (*Barry et al., 2007*; *Stensola et al., 2012*) combined unless otherwise noted. All error bars denote mean ± SEM. All reported statistics are paired t-tests, unless otherwise noted. (A) Whole-trial versus boundary-conditioned grid rescaling factors, normalized to range from no rescaling (0%) to a matched rescaling (100%). Boundary-conditioning resulted in a significant reduction in the appearance of rescaling (t(194) = 9.54, p<0.001; *Figure 3—source data 1*). (B) Field length along deformed and undeformed dimensions during compression deformations. (1st familiar vs. deformed: t(80) = 3.70, p<0.001; undeformed vs. deformed: t(86) = 2.43, p=0.017; 2nd familiar vs. deformed: t(82) = 3.49, p<0.001; all other comparisons: t < 1.45, p>0.151; *Figure 3—source data 2*). (C) Change in firing rates across conditions. (Peak firing rates, 1st familiar vs. deformation: t(80) = 3.57, p<0.001; 2nd familiar vs. deformation: t(82) = 3.34, p=0.001; 1st familiar vs. 2nd familiar: t(76) = 0.91, p=0.364; Mean firing rates, 1st familiar vs. deformation: t(80) = 3.57, p<0.001; 2nd familiar vs. deformation: t(82) = 3.34, p=0.001; 1st familiar vs. 2nd familiar: t(76) = 0.91, p=0.364; Mean firing rates, 1st familiar vs. deformation: t(80) = 0.54, p=0.591; 2nd familiar vs. deformation: t(82) = 0.03, p=0.978; 1st familiar vs. 2nd familiar: t(76) = 0.71, p=0.479; *Figure 3—source data 3*). (D) Examples of recorded and predicted rate maps during one deformation trial for two simultaneously recorded cell from (*Stensola et al., 2012*). (E) Correlation values between the recorded rate map and the rate maps predicted by the boundary-tethered model versus a matched rescaling during compression deformations (*Figure 3—source data 4*). (F) Cumulative distribution of the correlation values depicted in (E). The boundary-tethered model results in fewer low-similarity predictions than a matched rescaling (two-sample Kolmogorov-Smirnov test: D = 0.2030, p=0.007: *Figure 3—source data 4*). *p<0.05, **p<0.01, ***p<0.001.

DOI: https://doi.org/10.7554/eLife.38169.007

The following source data and figure supplement are available for figure 3:

**Source data 1.** Results of rescaling analysis.
DOI: https://doi.org/10.7554/eLife.38169.009
**Source data 2.** Results of field length analysis.
DOI: https://doi.org/10.7554/eLife.38169.010
**Source data 3.** Results of firing rate analysis.
DOI: https://doi.org/10.7554/eLife.38169.011
**Source data 4.** Results of map prediction analysis.
DOI: https://doi.org/10.7554/eLife.38169.012
**Figure supplement 1.** Predicting whole-trial rate maps from boundary-tethered shifts in grid phase.
DOI: https://doi.org/10.7554/eLife.38169.008

portion of the familiar rate map was used to construct the boundary rate map. Next, each boundary rate map was weighted by the actual sampling biases of the rat during that deformation trial. The final boundary-tethered prediction was then the smoothed sum of these weighted predicted boundary rate maps (See Materials and methods). For comparison, we also computed a rescaled rate map in which the familiar rate map was rescaled to match the deformation. Because additional fields may appear during stretching deformations which were not sampled in the smaller familiar environment, we focused only on compression trials. Across cells, recorded rate maps were more similar to those predicted by the boundary-tethered model than to those predicted by a matched rescaling (*Figure 3D*; *Figure 3—figure supplement 1*; *Figure 3—source data 4*), as quantified by the correlations between maps (paired t-test comparing Fisher-transformed correlation values: t(132) = 2.95, p=0.004; *Figure 3E*). This difference was predominantly driven by cells whose activity did not resemble a matched rescaling: recorded rate maps which were well-predicted by a matched rescaling were often similarly well-predicted by the boundary-tethered model, while recorded maps which were not well-predicted by a matched rescaling were nevertheless well-predicted by the boundary-tethered model. This pattern was reflected in the observation of fewer low-similarity predictions from the boundary-tethered model than from a matched rescaling (*Figure 3F*). We note, however, that a subset of recorded rate maps were better predicted by a matched rescaling, suggesting that the contribution of boundary-tethered phase shifts to whole-trial rate map distortions may vary (*Figure 3E*). Nevertheless, boundary-tethered phase shifts can accurately predict individual whole-trial rate maps on a trial-by-trial basis, even when the resulting rate map does not resemble a rescaling.

In sum, we have shown that conditioning grid cell activity according to the most recently contacted boundary during environmental deformations reveals grid patterns which are shifted relative to one another, anchored to the conditioned boundary, and appear less rescaled than the whole-trial grid pattern. Furthermore, we have shown that whole-trial field length increases along deformed dimensions, and whole-trial peak firing rates decrease during deformations while mean firing rate remains unchanged, matching boundary-tethered predictions. Finally, we have demonstrated that the boundary-tethered phase shifts, coupled with the particular sampling of the navigator, can accurately predict whole-trial rate maps during deformations whether or not the resulting maps resemble a matched rescaling. Together, these results provide convergent evidence that boundary-tethered shifts in grid phase are present during environmental deformations and contribute to distortions of the time-averaged grid pattern.

## A model of border cell-grid cell interactions reproduces boundary-tethered phase shifts, scale-dependent rescaling, and local distortions during deformations

Electrophysiological experiments have shown that rescaling a familiar environment can induce a corresponding rescaling of time-averaged grid patterns, dependent on grid scale (*Barry et al., 2007*; *Stensola et al., 2012*). Relatedly, when part of one wall is displaced, the time-averaged grid pattern is locally distorted near that wall, with grid fields neighboring that wall shifting in the direction of displacement (*Krupic et al., 2018*). Our reanalysis has demonstrated that boundary-tethered shifts in grid phase are observed during rescaling deformations and contribute to the appearance of rescaling in the time-averaged rate maps of grid cells. Together, these results suggest that boundary-tethered phase shifts may also contribute to the scale-dependent and local nature of these deformation-induced distortions.

We addressed this possibility computationally by implementing a spiking network model of the interactions between border and grid cells which captured boundary-tethered phase shift dynamics (*Keinath, 2018*). (To disambiguate recorded from modeled border and grid cells, we refer to all modeled cells as units.) The border population consisted of 32 units whose activity was designed to mimic the behavior of border cells (*Solstad et al., 2008*). Each border unit was active only when a boundary was nearby, within 12 cm in a particular allocentric direction (*Hardcastle et al., 2015*). The preferred firing field of each border unit covered 50% of the length of one side of the environment. Across units, fields were uniformly distributed around the perimeter with the fields of some units wrapping around corners (*Figure 4A*). Fields maintained proportional coverage when the preferred boundary was deformed (*Pollock et al., 2018*; *Solstad et al., 2008*; *Savelli et al., 2008*). The model also instantiated five grid modules, each consisting of a neural sheet of size 128 × 128 units. The

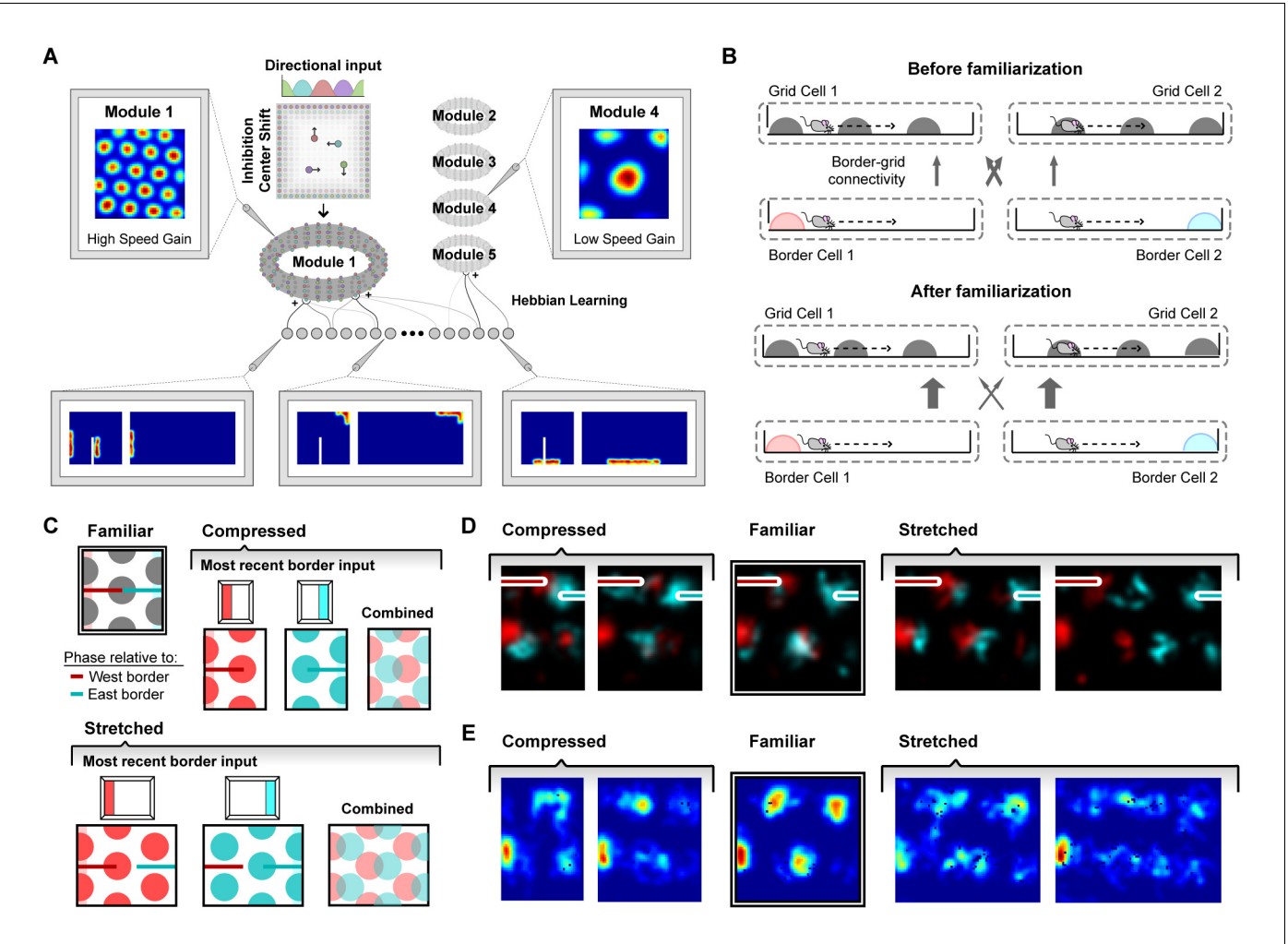

**Figure 4.** A model of border cell-grid cell interactions reproduces boundary-tethered shifts in grid phase during environmental deformations. (A) The network model consisted of two layers: a border layer, where unit activity was determined by the presence of a boundary nearby (<12 cm) and in a particular allocentric direction, and a grid layer, where path integration implemented by a periodic attractor network of the form described in *Burak and Fiete (2009)* was used to generate five modules of grid units of different scales. During deformations, border fields scale in concert with changes to their preferred boundary and shift when their preferred boundary is displaced. (B) During familiarization, excitatory connections from border units to coactive grid units were strengthened at the expense of non-coactive connections. In deformed environments, border inputs reinstated the familiar phase relative to the most recently contacted boundary (schematically (C) and in the model (D)). (E) Averaged over time, these shifts in grid phase can resemble rescaling.

DOI: https://doi.org/10.7554/eLife.38169.013

internal connectivity and dynamics of each module was based on an attractor model implementing path integration described in *Burak and Fiete (2009)*, and was identical across modules except for a single movement velocity gain parameter which controlled the grid scale of each module. This parameter was adjusted to yield a geometric series of scales across modules (scale factor of 1.42), as observed experimentally (*Stensola et al., 2012*) and explained theoretically (*Wei et al., 2015*; *Mathis et al., 2012*). In the absence of border unit input, grid unit activity in this model is determined by path integration alone.

In addition to the connections among units within its module, each grid unit also received initially random excitatory input from all border units. During familiarization, these connections developed through experience via a Hebbian learning rule in which connections between coactive grid and border units were strengthened at the expense of connections from inactive border units (*Grossberg, 1980*) (*Figure 4B*; see Materials and methods). Once familiarized, border unit input acted to

reinstate the grid network state associated with the same pattern of border unit responses during familiarization. This grid reinstatement occurred even when border inputs were activated at a new location, such as when a displaced boundary was encountered. In a rescaled environment, this grid reinstatement led to shifts in grid phase, such that the phase relative to the most recent border input matched the phase entrained during familiarization in the undeformed environment (*Figure 4C,D*). Averaged over time, these shifts in grid phase can resemble a rescaling (*Figure 4E*). Thus, deformations of a familiar environment induced boundary-tethered phase shifts in modeled grid units which can contribute to the appearance of rescaling, similar to those we observed in recorded grid cells.

Next, we tested whether the appearance of rescaling in this model depends on grid scale. To do so, we first familiarized a naive virtual rat with a 150 cm x 150 cm square environment. During this familiarization period, the border-grid connectivity self-organized via Hebbian learning. The virtual rat then explored the familiar environment and deformed versions of this environment without new learning (chamber lengths between 100 cm to 200 cm in increments of 25 cm; chamber sizes chosen to match experiment (*Stensola et al., 2012*)). Consistent with previous experimental reports (*Barry et al., 2007*; *Stensola et al., 2012*), we observed that these deformations induced rescaling of time-averaged rate maps in some grid modules (*Figure 5A*). To quantify this module-dependent rescaling, we computed the grid rescaling factor required to stretch or compress the time-averaged rate maps in the familiar environment to best match the rate maps in the deformed environment, separately for each unit. We found that the grid patterns of units in large-scale modules rescaled with the environment, but grid patterns of units in small-scale modules tended not to rescale (*Figure 5B*), as observed experimentally (*Stensola et al., 2012*). Rate-based simulations with larger border fields covering 100% of the length of one wall, and simulation in which boundary-tethered shifts were induced by fiat both reproduced these results (*Figure 5—figure supplement 1*; *Figure 5—figure supplement 2*).

How does scale-dependent rescaling emerge from boundary-tethered shifts in grid phase? Because the grid representation is periodic, border input can only reset the network state to within one period, analogous to a modulo operation. Generally, if the deformation extent is less than the grid period then different boundaries will reinstate different phases, resulting in a time-averaged grid pattern which appears rescaled. When the deformation extent nearly matches the grid period, different boundaries will reinstate a similar phase, yielding a largely undistorted time-averaged pattern. When the deformation extent exceeds the period, different boundaries will again reinstate different phases; thus, the time-averaged pattern will appear distorted. However, in the latter case, additional fields will appear (during stretches) or previously-observed fields will disappear (during compressions). Therefore, the time-averaged pattern, although distorted, will not resemble a simple rescaling of the grid to match the deformation.

In addition to capturing scale-dependent rescaling, this model further predicts that whether or not the grid patterns of a particular module appear to rescale is not an inherent property of the module, but rather an interaction between the grid scale and deformation extent. Specifically, modules with periods less than or equal to the deformation extent will tend not to resemble a rescaling, consistent with the data in (*Stensola et al., 2012*). Moreover, this model predicts that a grid with a given scale can appear to rescale during less extreme but not during more extreme deformations, consistent with comparison across experiments (*Barry et al., 2007*; *Stensola et al., 2012*; *Savelli et al., 2008*) (*Figure 5—figure supplement 3*). Finally, this model demonstrates that a single mechanism is capable of simultaneously reproducing the appearance and absence of rescaling across modules, suggesting that the extent of rescaling may not be clear evidence of a functional dissociation between modules (*Stensola et al., 2012*).

Lastly, we tested whether the grid patterns of modeled grid units locally distorted during partial deformations. We familiarized a naive virtual rat with either a 180 cm×90 cm rectangular or right trapezoid environment (long parallel wall of the right trapezoidal environment was 180 cm, short parallel wall was 135 cm). During this familiarization period, the border-grid connectivity self-organized via Hebbian learning. Without new learning, the rat then explored both the rectangular and right trapezoid environments. During deformations, fields near the displaced wall were distorted, often shifting in concert with the displaced wall, while fields far from this wall were less affected (*Figure 5C*). To quantify this pattern, we computed the correlation between the familiar and deformed environment rate maps across the population at each location, sometimes called the population vector correlation. This correlation was high at locations far from the displaced wall but was

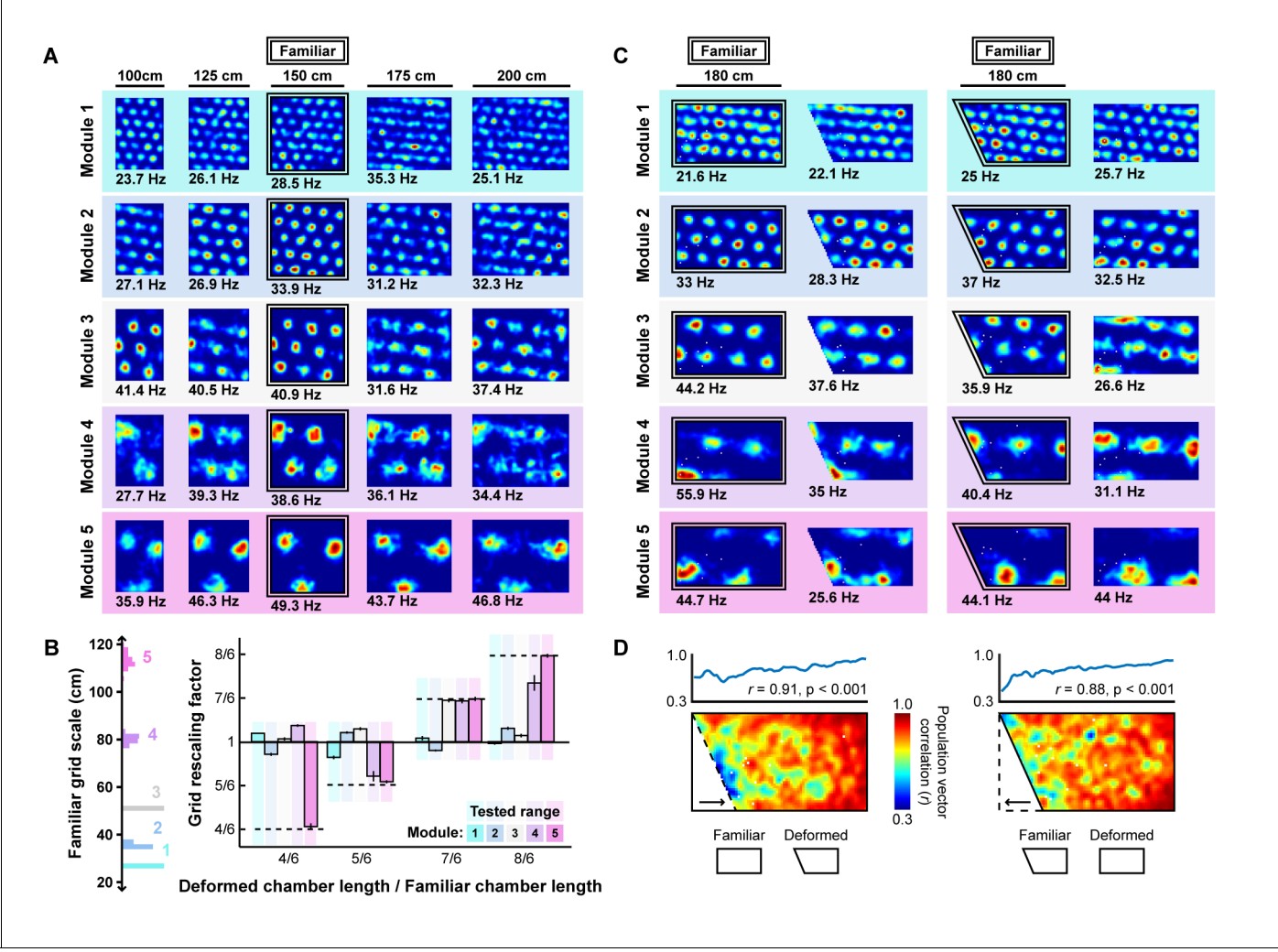

**Figure 5.** Grid unit responses to deformations of an open environment. (**A**) Rate maps from one grid unit from each module across all rescaling deformations. Colors normalized to the maximum across each set of rate maps. Peak firing rate for each trial noted below the lower left corner of each map. (**B**) Grid rescaling factors for each module when the familiar open environment is rescaled to various chamber lengths (right). Error bars denote standard error of the mean (SEM) across 30 random grid units. Color denotes module. Distribution of grid scales for each module indicated (left). (**C**) Rate maps of one grid unit from each module across each partial deformation, plotted as in (**A**). (**D**) Correlation between the familiar and deformed environment rate maps across the population (150 grid units, 30 random cells from each module) at each location (bottom heatmaps) and averaged at each east-west position (top plots).

DOI: https://doi.org/10.7554/eLife.38169.014

The following figure supplements are available for figure 5:

**Figure supplement 1.** A rate-based network model with full-length border units reproduces boundary-tethered shifts, scale-dependent grid rescaling, and local distortions of the grid pattern during deformations.

DOI: https://doi.org/10.7554/eLife.38169.015

**Figure supplement 2.** Scale-dependent rescaling arises from boundary-tethered shifts in simulations without plasticity.

DOI: https://doi.org/10.7554/eLife.38169.016

**Figure supplement 3.** A more extreme compression deformation does not produce matched rescaling.

DOI: https://doi.org/10.7554/eLife.38169.017

reduced near the displaced wall (*Figure 5D*), indicating that time-averaged grid patterns were distorted near the displaced wall but preserved further away from that wall, as observed experimentally (*Krupic et al., 2018*). Rate-based simulations with larger border fields covering 100% of the length of one wall also reproduced these results. (*Figure 5—figure supplement 1*).

How do local distortions emerge from nonlocal boundary-tethered shifts in grid phase during deformations? In the model, distortions of the whole-trial rate map arise from averaging over shifts in grid phase which occur following contact with displaced boundaries. Importantly, the likelihood of having most recently encountered a given boundary differs throughout an open environment: locations near a boundary are more likely to be visited following an encounter with that boundary, while central locations are less biased (*Figure 6A*). Because of these biases, time-averaged grid fields near a boundary will appear less distorted than central fields during stretching and compression deformations. Similarly, during partial deformations, locations near the displaced wall are more likely to be visited following contact with that wall; thus shifts in phase following contact will predominantly affect nearby grid fields, with the phase relationship between this wall and neighboring fields better preserved even after averaging over time. Relatedly, the most recently encountered boundary is also correlated with the direction of movement – a given boundary is more likely to have been recently encountered when moving away from that boundary (*Figure 6B*). Thus, boundary-tethered shifts in grid phase are also predicted to resemble modulation by movement direction, analogous to similar modulation observed in place cells (*O'Keefe and Burgess, 1996*). Thus, in this model sampling biases, a product of the particular path of the navigator, play a critical role in mediating the contribution of boundary-tethered shifts to distortions of the time-averaged grid pattern.

## Discussion

Our results support two primary conclusions. First, boundary-tethered shifts in grid phase are observed directly in the activity of recorded grid cells during environmental deformations, and contribute to the appearance of rescaling. Second, boundary-tethered shifts in grid phase, scale-dependent rescaling, and local distortions of the time-averaged grid pattern can emerge from border cell-grid cell interactions. Together, these results highlight previously unrecognized dynamics governing the grid code during environmental deformations and implicate border cell-grid cell interactions as a potential source of the dynamics we observe as well as of deformation-induced distortions of the grid pattern. These results further indicate that time-averaged analyses may have overestimated the malleability of the grid cell spatial metric in response to environmental deformations and suggest that scale-dependent grid rescaling may not be a clear indicator of a functional dissociation between modules. Finally, these results demonstrate that the effects of environmental deformations are not fixed over time, but instead depend crucially on the movement history of the navigator.

Through our simulations, we have demonstrated that simple interactions between border cells and grid cells are sufficient to reproduce boundary-tethered shifts in grid phase. However, a variety

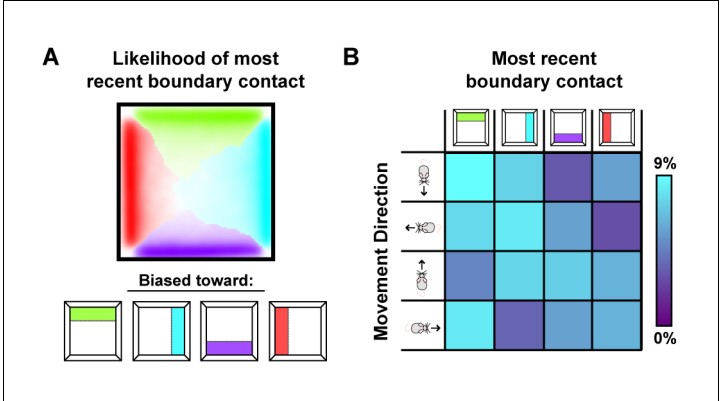

**Figure 6.** Sampling biases are correlated with the most recently contacted boundary. (**A**) Likelihood of having most recently contacted each border as a function of location in a square environment. Hue indicates the most likely recently contacted boundary; saturation denotes the strength of the bias (white – 25% likelihood of sampling; fully saturated – 100% likelihood of sampling). Data from (*Stensola et al., 2012*). (**B**) Joint probability distribution depicting the relationship between movement direction and the most recently contacted boundary. Data from (*Stensola et al., 2012*).
DOI: https://doi.org/10.7554/eLife.38169.018

of alternative circuits and cues could also give rise to boundary-tethered phase shifts. For example, visual inputs (*Pérez-Escobar et al., 2016*) and reciprocal connections from place to grid cells (*Bonnevie et al., 2013*) both play important roles in developing and maintaining a functional grid code, and may contribute to phase anchoring during deformations (*Sheynikhovich et al., 2009*). Moreover, similar boundary-tethered dynamics can be observed in the place code even before the grid code has fully matured, suggesting that additional mechanisms may contribute to similar dynamics in place cells (*Bjerknes et al., 2018*). Thus, while our results implicate border cell-grid cell interactions as one possible source of the experimentally observed grid phase shifts, additional experiments are required to causally test the particular circuit realization which gives rise to these shifts.

The dynamic boundary-tethered phase anchoring we observe here may reflect a more general phenomenon of grid phase anchoring to external cues or other internal reference frames (*Fuhs and Touretzky, 2006*; *Evans et al., 2016*). Consistent with this idea, the grid representation is shaped by a number of boundary and non-boundary cues even in geometrically undeformed environments. For example, grid scale differs between novel and familiar environments (*Barry et al., 2012*), the grid pattern is anchored both by spatial geometry and other visual features (*Savelli et al., 2017*; *Krupic et al., 2015*), and the grid pattern distorts near familiar boundaries as well as in asymmetric environments (*Krupic et al., 2015*; *Stensola et al., 2015*). These effects were not captured by the border cell-grid cell interactions as implemented here, and may reflect phase-anchoring to external cues (*Fuhs and Touretzky, 2006*; *Savelli et al., 2017*; *Stensola et al., 2015*) or internal reference frames such as those supported by boundary vector cells (*Bush and Burgess, 2014*; *Krupic et al., 2014*) or place cells (*Evans et al., 2016*; *Bush and Burgess, 2014*).

Our results do not rule out additional dynamics and mechanisms which may be at play during environmental deformations. Indeed, it is likely that multiple mechanisms contribute to the various properties of deformation-induced grid distortions. For example, it is known that during deformations the distorted grid pattern does not persist indefinitely, but relaxes back to the familiar spatial scale with experience (*Barry et al., 2007*). However, in our simulations, model weights were fixed during deformation trials so that no further learning occurred in this phase. Indeed, the effects we observe can be reproduced by directly imposing boundary-tethered grid shifts without invoking plasticity (*Figure 5—figure supplement 2*). Even with continued learning, the model as implemented here may not capture long-term relaxation dynamics because grid phase and border input are likely not in conflict long enough for unlearning to occur. More specifically, when the west boundary is encountered following an east boundary contact during an east-west deformation, the border and grid codes are briefly in conflict when the border representation is first activated, causing a small amount of unlearning. However, this border activation also quickly reinstates the learned grid phase, eliminating the conflict between the two. The learned grid phase is then reinforced for as long as the animal remains close to the west boundary, which will typically be long enough to overwrite whatever unlearning has occurred. Thus, other mechanisms, such as anchoring to additional conflicting reference frames (input from visual cues [*Fuhs and Touretzky, 2006*; *Sheynikhovich et al., 2009*; *Bjerknes et al., 2018*; *Raudies and Hasselmo, 2015*], boundary vector cells (*Barry et al., 2006*; *Stewart et al., 2014*), or place-to-grid feedback (*Bush and Burgess, 2014*]) or changes to speed coding (*Zilli, 2012*), are necessary to explain grid relaxation.

Previous work has also revealed conspicuous parallels between deformation-induced distortions of grid and place cell representations in the rat brain and the spatial memory of humans in deformed environments (*O'Keefe and Burgess, 1996*; *Hartley et al., 2004*; *Hartley et al., 2000*; *Chen et al., 2015*), leading to the suggestion that a common mechanism might underlie these effects. Consistent with this idea, recent evidence suggests that rescaling can be observed in the time-averaged activity of human grid cells (*Nadasdy et al., 2017*). In light of our results and further simulations of place cell activity (see Materials and methods), we suggest that boundary-tethered grid shift may be a common mechanism contributing to these cross-species effects, and predict that similar boundary-tethered shifts should be observed in human spatial memory during environmental deformations.

# Materials and methods

## Model

### Border layer

The border layer consisted of 32 units. First, the area near each wall in 4 allocentric directions (North, South, East, West) was divided into 8 'bricks' (see [22] for a similar treatment). Each brick extended 12 cm perpendicular from the wall and covered 12.5% of the total environment length along that dimension. Each unit $j$ received a uniform input $b_j = 0.1$ whenever the simulated rat was within one of four adjacent bricks, resulting in a firing field covering 50% of the environment perimeter for each unit. This input was converted to stochastic spiking activity (see below).

### Grid layer

The grid layer, derived from the model of [30], consisted of five grid 'modules'. Each module consisted of a neural sheet with periodic boundary conditions, visualized as a torus. This neural sheet was composed of $64^2$ identical 2 unit x 2 unit tiles ($128^2$ units per module). Each unit in a tile was associated with a particular direction (North, South, East, West), which determined both the movement-direction-specific excitatory input received, as well as its local connectivity. Movement-direction-specific excitatory input $v_j$ to grid unit $j$ was determined by

$$v_j = \gamma + g_m\big(d\cos(\theta - \phi_j)\big)$$

where $d$ is the distance moved since the previous timestep, $\theta$ is the direction of movement, $\phi_j$ is the preferred direction of unit $j$, $g_m$ is a gain factor specific to the module $m$ to which to unit $j$ belongs, and $\gamma = 0.6$ is a constant. Local connections within each module consisted of shifted radial inhibition, in which each unit inhibited all units within a 12 unit radius by a uniform weight of -0.02. The center of this radial inhibition output for each unit was shifted by 2 units away from that unit in a direction consistent with each units preferred direction. In the absence of other inputs, each grid module yields a hexagonal grid-like pattern of activation on the neural sheet, which is translated during movement at a rate proportional to the gain factor. Thus, to model modules with varying grid scales, the gain factor $g_m$ of module $m$ was set by

$$g_m = \frac{g_1}{2^{\left(\frac{m-1}{2}\right)}}$$

where $g_1 = 0.45$ is the gain of the smallest-scale module, module 1. This results in a geometric series of biologically-plausible (*Stensola et al., 2012*) grid scales for each module.

### Place layer

The place layer consisted of 64 units, subject to uniform recurrent inhibition from all place layer units with a weight of $-0.15$.

### Border-to-grid connectivity

All grid units received additional excitatory feed-forward projections from all border units. These connections were initialized with random weights uniformly sampled from the range 0 to 0.025, and developed through experience via Hebbian learning (see below and (*Pollock et al., 2018*)).

### Grid-to-place connectivity

Each place unit received additional excitatory feed-forward projections from 500 random grid units. These connections were initialized with random weights uniformly sampled from the range 0 to 0.022, and developed through experience via Hebbian learning (see below).

## Model dynamics

### Activation

The dynamics of the network was developed following the methods in [30]. The activation $a_j$ of unit $j$ was determined by first computing the total input $b_j$ to unit $j$ according to

$$b_j = \begin{cases} v_j + \sum_i^I a_i w_{ij}, & \text{grid units} \\ \sum_i^I a_i w_{ij}, & \text{place units} \end{cases}$$

where $a_i$ is a variable quantifying activation of unit i, $w_{ij}$ is the weight from unit i to unit j, and Ienumerates all the units. (Note that some weights $w_{ij}$ can be zero.) Also recall from above that a border unit receives a constant input when the rat is in a boundary region associated with that unit. The total input $b_j$ was used to stochastically determine the spiking $s_j$ of each unit j during the current timestep, according to

$$s_j = \begin{cases} 1, & \kappa(b_j - \beta_j)dt > \text{unif}(0,1) \\ 0, & \kappa(b_j - \beta_j)dt \leq \text{unif}(0,1) \end{cases}$$

where $\kappa = 500$ is a scale factor, $\beta_j$ (border units: $\beta_j = 0$; grid units: $\beta_j = 0.1$; place units: $\beta_j = 0.05$) is the spike threshold for unit j, $\text{unif}(0,1)$ is a single draw from a random uniform distribution ranging from 0 to 1, and $dt = 0.003$ sec is the length of each timestep. Finally, this spiking activity was integrated to update the activation variable $a_j$ of unit jafter each timestep according to

$$a_j = a_j - a_j \frac{dt}{c} + \alpha s_j$$

Where $\alpha = 0.5$ is a scale factor and $c = 0.03$ sec is the time constant of integration.

## Hebbian learning
All Hebbian weigh ts were updated by the competitive learning rule

$$w_{ij} = w_{ij} + \lambda a_j \left( \left( (\xi_j - w_{ij}) a_i \right) - \left( w_{ij} \sum_{n \neq i} a_n \right) \right)$$

where the sum is only over the set of units with nonzero Hebbian weights to unit j, $\lambda = 0.00001$ is the learning rate, $\xi_j$ is a constant specific to the connection type (border-to-grid: $\xi = 0.4$; grid-to-place: $\xi = 0.5$) (*Grossberg, 1980*; *Pilly and Grossberg, 2013*). This rule results in competitive activity-dependent weight changes among incoming Hebbian connections, and leads over time to a total weight of $\xi_j$ across incoming synapses.

## Simulation details
### Generating simulated rat paths
Because some of the deformed environments that we tested have not been experimentally studied, it was necessary to generate simulated rat paths, rather than using experimentally recorded paths. Open-field paths were generated via a bounded random walk model, parameterized by speed and movement direction. At each timestep, unbiased normally distributed random noise was added to both speed ($\sigma = 0.001$ cm/msec) and movement direction ($\sigma = 1.5$ °/ms). To approximate actual rat exploration, speed was bounded to the range [0, 40] cm/s. If a step would result in the rat path crossing a boundary, random noise was again added repeatedly to the movement direction until the next step would no longer cross the boundary. Open field paths always began in the center of the environment, with the simulated rat stationary and facing a random direction. Linear track paths were generated as straight end to end laps at a constant speed of 20 cm/s.

### Familiarization
In all simulations, familiarization with the environment was mimicked by allowing the naive simulated rat to explore the environment for 60 min. Prior to familiarization, grid layer activity was allowed to settle into its grid-like attractor state for 2 s without learning. Initialization of the grid layer was biased so that an axis of the settled grid network state would lie at an angle of $-7.5°$ relative to east, consistent with experiments (*Krupic et al., 2015*; *Stensola et al., 2015*). Following

familiarization, the model weights were saved so that all post-familiarization simulations could begin with the familiarized model.

## Post-familiarization testing simulations

The model weights were reset to the state saved after familiarization, and the experienced virtual rat was allowed to explore each tested environment for 30 min. Grid layer activity was also initially reset to the familiar environment state corresponding to the rat's start location. Learning was turned off during the testing phase.

## **Analysis**

### Statistical tests

All statistical tests are two-tailed unless otherwise noted. All error bars denote mean ±1 standard error of the mean unless otherwise noted.

### Reanalysis of experimental data

A complete description of the experiments was provided in (*Barry et al., 2007*; *Stensola et al., 2012*). Data from *Barry et al., 2007*) included an initial set of 66 putative cells, from which 38 cells meeting various criteria were selected as grid cells for analysis in the original publication. Similarly, we included only cells with average gridness across both familiar trials > 0.4 from this dataset, yielding 36 included grid cells. Note that unlike in *Barry et al., 2007*) we did not exclude cells which were poorly fit by rescaling during deformation trials, as boundary-tethered phase shifts predict that distortions which do not resemble a rescaling may occur. For alignment, rescaling, and rate map prediction analyses, the first familiar trial rate map were used for comparison; in the few cases where no rate map was recorded during the first familiar trial, the rate map from the second familiar trial was used instead.

### Unit sampling

Due to computational constraints and the redundant nature of grid unit activity, only the spikes from 30 randomly chosen grid units in each module were recorded and analyzed during all simulations. All place units were recorded and analyzed.

### Rate maps

Rate maps were created by first dividing the environment into 2.5 cm x 2.5 cm pixels. Then the mean firing rate within each pixel was calculated. Finally, this map was smoothed with an isotropic Gaussian kernel with a standard deviation of 1.5 pixels (3.75 cm) and square extent of 9 pixels x 9 pixels (22.5 cm x 22.5 cm). Pixels which were never visited were excluded during further analyses, with the exception of rate map prediction: all pixels were included during rate map prediction as even few missing pixels lead to large gaps of missing pixels following rescaling.

### Autocorrelations and cross-correlations

Autocorrelations of rate maps were computed similar to previous reports (*Sargolini et al., 2006*). Briefly, the correlation $r$ between overlapping pixels of the original rate map and a shifted version of itself was computed as

$$r = \frac{\sum_{i=1}^{I}\sum_{j=1}^{J}\left(f_{ij}-\bar{f}\right)\left(f'_{ij}-\bar{f'}\right)}{\sqrt{\sum_{i=1}^{I}\sum_{j=1}^{J}\left(f_{ij}-\bar{f}\right)^2}\sqrt{\sum_{i=1}^{I}\sum_{j=1}^{J}\left(f'_{ij}-\bar{f'}\right)^2}}$$

where $f$ is the rescaled rate map, $f'$ is the familiar rate map, $i$ and $j$ run over pixels in the overlapping regions of these maps, and $\bar{f}$ and $\bar{f'}$ indicate the mean firing rate across overlapping pixels, at a series of single pixel (2.5 cm) step lags. Cross-correlations were computed similarly, except that two different rate maps were correlated, rather than two copies of the same rate map. Autocorrelations and cross-correlations were only estimated for spatial lags with at least 20 overlapping pixels.

## Grid scale

To compute grid scale for a unit or cell we first computed the rate map autocorrelation. Next, we computed the mean distance from the center of the autocorrelation to the center of mass of the six closest surrounding peaks, excluding the central peak.

## Gridness

To compute gridness for each unit, we first computed the autocorrelation of its rate map and its grid scale. Next we masked the autocorrelation, eliminating all pixels at a distance from the center greater than 1.5 its scale and less than 0.5 its scale. We then computed the correlation between the masked autocorrelation and a rotated version of itself, rotated 30°, 60°, 90°, 120°, and 150°. The final measure of gridness was then the difference between the minimum of the [60° 120°] correlations minus the maximum of the [30° 90° 150°] correlations.

## Field length

Field length along each dimension was estimated from the autocorrelation by first determining the extent of the central peak of the autocorrelation, defined as all contiguous pixels with correlation values greater than 10% of the maximum correlation. Next, field length was computed separately for each dimension as the distance between the most extreme pixels within this central peak along that dimension.

## Grid rescaling factor

The grid rescaling factor during each deformation trial was computed separately for each unit by comparing rescaled versions of the familiar environment rate map to the deformed environment rate map. Following (**Stensola et al., 2012**), the familiar rate map was uniformly rescaled to a series of chamber lengths, ranging from 10 cm below the smaller of the deformed and familiar chamber lengths, through 10 cm above the larger of these chamber lengths in 5 cm (2 pixel) increments. This yielded a set of rescaled familiar rate maps for each unit. For each rescaled map, we computed the correlation $g_1 = 0.56$ (defined above) between the deformed and rescaled rate maps twice, once when the two rate maps were aligned by each opposing boundary. The grid rescaling factor was then defined as the ratio between the rescaled chamber length that yielded the highest correlation and the familiar chamber length, across either alignment.

When comparing rescaling factors between whole-trial and boundary-conditioned data, rescaling was only computed for alignment by the conditioned boundary. In order to ensure that differences in rescaling following boundary-conditioning were not due to differential sampling following contact with opposing boundaries, whole-trial rate maps were created from random subsets of the data, with the number of samples at each pixel chosen to match the sampling distribution of the to-be-compared boundary rate map. Because subsampling the data to match these sampling distributions introduces variability in the measure of rescaling, we repeated this measure 100 times for each cell with different random subsets of the data and took the mean rescaling across these iterations as our final measure.

## Grid shift analysis

To test these data for the presence of grid shifts during environmental deformations, we first divided the spiking activity of each cell according to the most recent boundary contact (North, South, East, or West). Boundary contact was defined as the rat being within 12 cm of a boundary. Spiking activity prior to boundary contact at the beginning of the trial was ignored. Next, four separate rate maps were created, one for each most recently contacted boundary. To quantify grid shift along a particular dimension for each cell, the rate maps of opposing boundaries perpendicular to the chosen dimension were cross-correlated at a series of lags in single pixel steps (see above) within the range of plus/minus one half the scale of the grid pattern. In order to ensure that shifts in phase were not due to differential sampling following contact with opposing boundaries, boundary rate maps were created from random subsets of the data, with the number of samples at each pixel chosen to match the sampling distributions of the opposing boundary rate maps. The distance from the center to the nearest peak of this cross-correlogram along the dimension of interest was then determined, with the nearest peak defined by first partitioning the cross-correlogram into 'blobs' of contiguous pixels

which had positive correlations. Then, the location with the maximum correlation value within the blob nearest to the center was taken as the nearest peak. Because subsampling the data to match the sampling distributions introduces variability in this measure of shift, we repeated this measure 100 times for each cell with different random subsets of the data and took the mean shift across these iterations as our final measure of shift. Because shift is bounded by grid scale, we report shift as a ratio of shift to scale.

## Boundary-tethered rate map prediction

For each cell and deformation trial we first created predicted boundary rate maps for displaced boundaries from the familiar environment rate map. These rate maps were shifted versions of the familiar rate map, aligned by the corresponding boundary (*Figure 3—figure supplement 1*). If the length of a boundary changed, then the central portion of the familiar rate map was used to generate the predicted boundary rate map. Next, sampling biases were applied as follows. First, a map of the actual sampling behavior following each boundary contact during the deformation trial was computed, as described in the 'rate maps' section above. From these maps, the probability of having most recently contacted each boundary was computed at each pixel. The contribution from each boundary rate map was then weighted by this probability. The final rate map predicted by the boundary-tethered model was then the sum of these weighted boundary rate maps, smoothed with the Gaussian kernel described in the 'Rate maps' section above.

## Data and code availability

All simulations were conducted with custom-written MATLAB scripts. These scripts as well as the simulation results presented here are available on Github at https://github.com/akeinath/Keinath_BoundaryTetheredModel (*Keinath, 2018*; copy archived at https://github.com/elifesciences-publications/Keinath_BoundaryTetheredModel). All values generated by our reanalysis are available as source data files. All original reanalyzed data were originally reported in the following papers:

- *Barry et al., 2007*. *Experience-dependent rescaling of entorhinal grids*. https://doi.org/10.1038/nn1905
- *Stensola et al., 2012*. *The entorhinal map is descritized*. https://doi.org/10.1038/nature11649 These data are available upon request from the corresponding authors of these papers.

## Simulations of place units

Environmental deformations also induce distortions of place cell activity which may reflect a contribution of dynamically-shifting grid input. To test this possibility, we concurrently simulated a population of place units receiving input from grid units during a variety of deformations.

Electrophysiological experiments have shown that stretching a familiar environment induces a heterogeneous mix of responses in the time-averaged activity of place cells (*O'Keefe and Burgess, 1996*). To explore the effects of stretching deformations on model place units, we began by familiarizing the naive virtual rat with a 61 cm×61 cm square open environment, during which period the border-grid connectivity and grid-place connectivity self-organized via Hebbian learning. Following this familiarization, the virtual rat again explored the familiar environment, as well as a number of deformed environments without new learning (various chamber lengths between 61 cm and 122 cm, chamber widths 61 cm or 122 cm; chamber sizes chosen to match experiment (*O'Keefe and Burgess, 1996*)). During these deformations, we observed heterogeneous changes in the time-averaged rate maps of place units (*Figure 7A*). A number of place units exhibited place field stretching in proportion to the rescaling deformation. Other units exhibited place field bifurcations accompanied by progressively lower peak firing rates during more extreme deformations. Finally, some units exhibited emergent modulation by movement direction, with place fields shifting 'upstream' of the movement direction. A qualitatively similar mix of place field distortions is observed experimentally (*O'Keefe and Burgess, 1996*).

Electrophysiological experiments have also demonstrated that when a familiar linear track is compressed, the place code is updated when track ends are encountered (*Gothard et al., 1996*; *Gothard et al., 2001*). We therefore examined the effects of compressing a familiar linear track on model place units. We first familiarized the naive virtual rat with running laps on a 161 cm long linear track, during which period the border-grid connectivity and grid-place connectivity self-organized

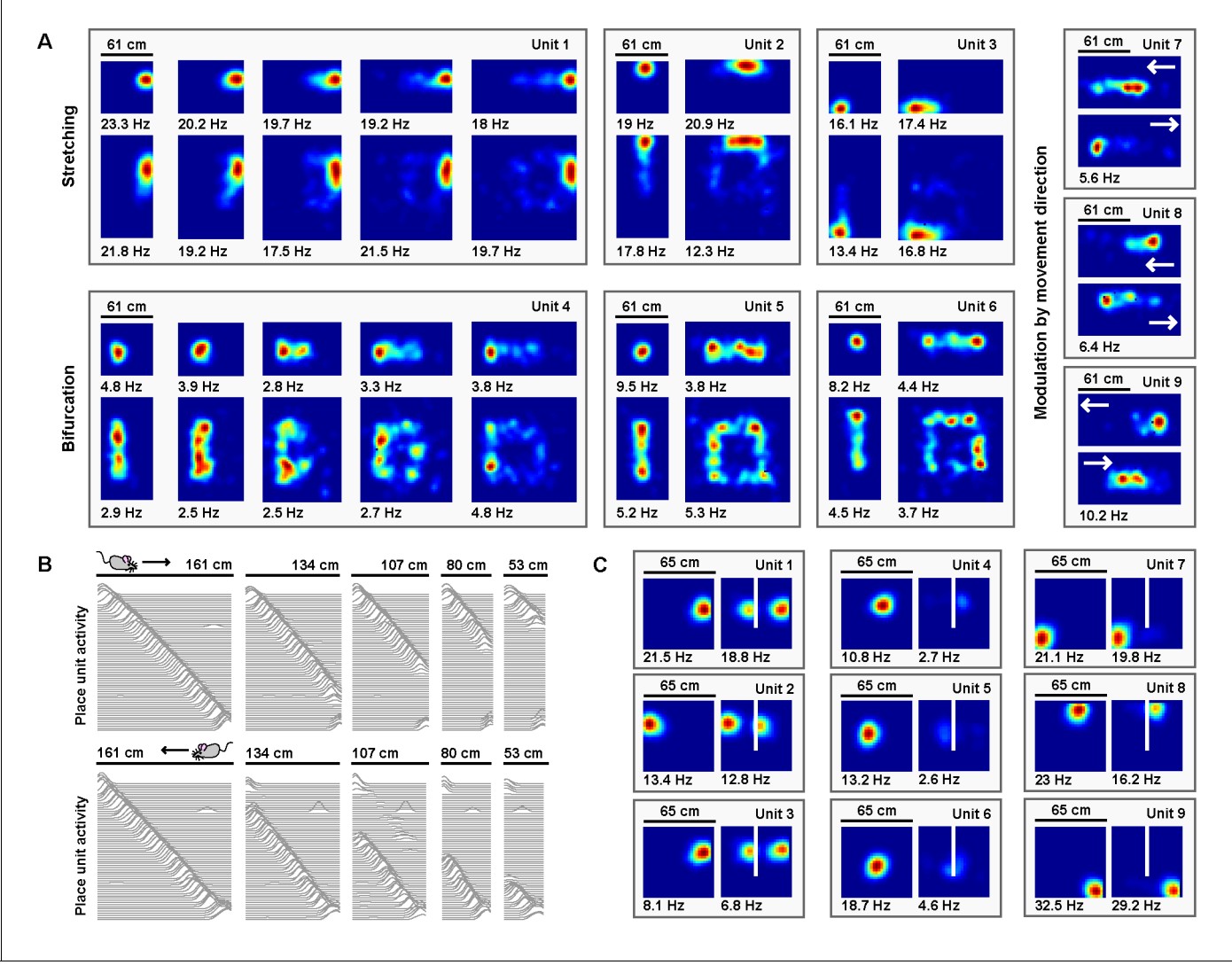

**Figure 7.** Place units learned from grid unit inputs reproduce heterogeneous place field distortions. (**A**) Place unit rate maps when a familiar open environment is stretched. Place fields exhibit stretching, bifurcation, and emergent modulation by movement direction (indicated by white arrows). Colors normalized to the peak for each rate map. Peak firing rate noted below the lower left corner of each map. Note that peak firing rate tends to decrease with more extreme deformations for cells with place fields further from boundaries. (**B**) Place unit activity for all 64 place units during compressions of a familiar linear track, separated by (top) eastward and (bottom) westward laps. Each line indicates the firing rate of a single place unit at each location across the entire track during movement in the specified direction, normalized to the familiar track peak rate. Units sorted by place field location on the familiar track. Note that, during compressions, the place code unfolds as if anchored to the beginning of the track until the end of the track is encountered, at which point the familiar end-of-track place units are reactivated. (**C**) Place unit rate maps demonstrating a mix of place field (left) duplication, (middle) inhibition, and (right) perseverance when a new boundary (white line) is inserted in a familiar open environment. Colors normalized to the maximum of both rate maps. Peak firing rate noted below the lower left corner of each map.

DOI: https://doi.org/10.7554/eLife.38169.019

via Hebbian learning. Border units were active only within 12 cm of each track end. Following this familiarization, the virtual rat ran laps along both the familiar track and a number of compressed tracks without new learning (track lengths between 53 cm to 161 cm; lengths chosen to match experiment (*Gothard et al., 1996*)). During laps on compressed tracks, place unit activity unfolded as if unaffected by the compression, no matter how extreme, until the opposing track end was reached. Once encountered, the place code previously active at this track end during familiarization reemerged (*Figure 7B*), as observed experimentally (*Gothard et al., 1996*). In recording experiments, similar boundary-tethered updating persists in darkness indicating that such dynamics arise

even in the absence of visual cues (*Gothard et al., 2001*), a result consistent with the sustained activity of border cells in darkness (*Pérez-Escobar et al., 2016*; *Chen et al., 2016*). However, we note that in these recording experiments the particular transition point differs depending on the availability of visual input and may precede border cell firing, which likely reflects the influence of additional mechanisms outside the scope of our boundary-tethered model (*Sheynikhovich et al., 2009*; *Raudies et al., 2016*).

Finally, electrophysiological experiments have shown that when a boundary is inserted in a familiar open environment, place fields exhibit a mix of duplication, suppression, and perseverance (*Barry et al., 2006*; *Muller and Kubie, 1987*; *Lever et al., 2002*). We explored the effects of inserting a new boundary on model place units. We first familiarized the naive virtual rat with a 65 cm×65 cm square open environment, during which period the border-grid connectivity and grid-place connectivity self-organized via Hebbian learning. Following this familiarization, the rat explored, without new learning, the familiar environment and a deformed version of this environment containing an additional 40 cm long boundary adjacent to one wall and evenly dividing the space (chosen to match experiment (*Barry et al., 2006*)). Again, we observed heterogeneous changes in the time-averaged rate maps of place units (*Figure 7C*). Some units exhibited place field duplication during boundary insertion, while other units exhibited place field inhibition. Still others persevered largely unaffected. A qualitatively similar mix of responses is observed experimentally during boundary insertions (*Barry et al., 2006*; *Muller and Kubie, 1987*; *Lever et al., 2002*). Together, these results demonstrate that many of the heterogeneous place cell behaviors observed across environmental deformations could arise from boundary-tethered shifts in grid code input.

## Acknowledgements

We are grateful to the laboratories of Edvard and May-Britt Moser, Kate Jeffery and Dori Derdikman for making the data from (*Stensola et al., 2012*) and (*Barry et al., 2007*) available for our reanalysis. We also thank Eli Pollock, Niral Desai and Xuexin Wei for advice on implementing spiking border-grid connections. Finally, we gratefully acknowledge support from NSF grant PHY-1734030 (VB), NIH grants EY022350 and EY027047 (RAE), and NSF IGERT grant 0966142 (ATK). VB was also partially supported by the Honda Research Institute Curious-Minded Machines program. VB thanks the Aspen Center for Physics (Aspen, Colorado; NSF grant PHY-1607611) and the International Center for Theoretical Physics (Trieste, Italy) for support and hospitality while this paper was being completed.

## Additional information

### Funding

| Funder | Grant reference number | Author |
| --- | --- | --- |
| National Science Foundation | PHY-1734030 | Vijay Balasubramanian |
| National Institutes of Health | EY022350 | Russell A Epstein |
| National Science Foundation | 0966142 | Alexandra T Keinath |
| National Institutes of Health | EY027047 | Russell A Epstein |
| National Science Foundation | PHY-1607611 | Vijay Balasubramanian |
| Honda Research Institute | Curious-Minded Machines program | Vijay Balasubramanian |

The funders had no role in study design, data collection and interpretation, or the decision to submit the work for publication.

### Author contributions

Alexandra T Keinath, Conceptualization, Data curation, Software, Formal analysis, Validation, Investigation, Visualization, Methodology, Writing—original draft, Writing—review and editing; Russell A Epstein, Resources, Supervision, Funding acquisition, Methodology, Writing—review and editing;

Vijay Balasubramanian, Conceptualization, Resources, Formal analysis, Supervision, Funding acquisition, Methodology, Project administration, Writing—review and editing

## Author ORCIDs
Alexandra T Keinath (ID) http://orcid.org/0000-0003-1622-7835
Vijay Balasubramanian (ID) https://orcid.org/0000-0002-6497-3819

## Decision letter and Author response
Decision letter https://doi.org/10.7554/eLife.38169.022
Author response https://doi.org/10.7554/eLife.38169.023

## Additional files

### Supplementary files
• Transparent reporting form
DOI: https://doi.org/10.7554/eLife.38169.020

All simulations were conducted with custom-written MATLAB scripts. These scripts as well as the simulation results presented here are available on Github at: https://github.com/akeinath/Keinath_BoundaryTetheredModel (copy archived at: https://github.com/elifesciences-publications/Keinath_BoundaryTetheredModel). All values generated during our reanalysis are included as source data files. All original reanalyzed data were originally reported in the following papers: 1) Barry et al., 2007. Experience-dependent rescaling of entorhinal grids. https://doi.org/10.1038/nn1905; 2) Stensola et al., 2012. The entorhinal map is descritized. https://doi.org/10.1038/nature11649. These data are available upon request from the corresponding authors of these papers.

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
