## [Decision Letter]

Thank you for submitting your article "Environmental deformations dynamically shift the grid cell spatial metric" for consideration by *eLife*. Your article has been reviewed by three peer reviewers, and the evaluation has been overseen by a Reviewing Editor and Michael Frank as the Senior Editor. The following individual involved in review of your submission has agreed to reveal his identity: Adrien Peyrache (Reviewer #2). Two other reviewers remain anonymous.

The reviewers have discussed the reviews with one another and the Reviewing Editor has drafted this decision to help you prepare a revised submission.

Summary:

Place cells and other spatial cells found in the hippocampal formation strongly suggest the existence of a cognitive map in this area. Grid cells seem to reflect the metrical properties of this map, presumably derived from path integration computations. One reason the metrical character of the grid representation is rejected by some scientists is the fact that the regular grid firing pattern is susceptible to distortions following several types of experimental manipulations. Data from some of these experiments are reanalyzed in this study according to the computational hypothesis that these distortions can be (at least in part) caused by different reset points along the boundaries of the environment. When the boundaries of a familiar environment are manipulated, these points can produce shifted grid representations depending on which point was last encountered by the animal. Averaging together these shifted maps into a whole-trial firing rate map will yield a highly distorted grid. Specifically, the main finding is that the time averaged spatial discharge of grid and place cells look distorted only because it results from the superposition of grid/place fields that are offset depending on the origin of the animal during a particular journey. These new analyses were guided by insightful computational modeling and suggest that grid cells firing patterns reflect a more consistent metric than appears from traditional analyses.

The reviewers agreed that this work is important. However, they raised some concerns, the most substantial of which perhaps relate to the presentation of results. The reviewers feel that the most novel and important parts of the paper are the analyses, which are perhaps not properly highlighted in the current version of the manuscript. The reviewers' other major concerns are detailed below.

Essential revisions:

1) The model is only weakly connected to the data analysis part. As the authors themselves admit in the Discussion, the specific design of the model is not very important (Discussion, second paragraph). The main theoretical idea that border/boundary information resets some path integrating mechanism is actually quite old (and was already concluded from the Gothard et al., 1996 data). This questions whether the present model has any added value, particularly since all model components are well-known in the literature and many of the model assumptions are highly debated (which is not even discussed in the paper). The model should be more clearly presented in the foundation of previous work. Also, the authors should present the results of their data analysis first, and then present the model as a potential mechanistic explanation of the results. In general, they should devote more of the main text to the analysis component (about 3/5 of the supplementary materials are devoted to this part).

2) The border fields in the model are smaller than typical, firing along only 50% of the border. Although this is not abnormally small, most border cells published in the literature appear to fire along longer segments of a boundary, often times along the entire boundary and extending across a corner. One wonders if the model still works if border cells are incorporated that simulate the range of lengths reported in the literature, with the majority firing along the entire border (or at least >= 75% of the border). Although the analysis of the data based on the authors' insights from the model is the real contribution of this paper, I think it is important for the authors to show whether the model still works with appropriately sized border cells (if for no other reason than to forestall distraction from the experimental analyses if readers are skeptical about the model). If the model fails to produce grid resets with larger border cell fields (perhaps due to ambiguities between the orientation of the grid fields and the extended borders), this would be an important point to discuss, suggesting that other inputs than border cells are necessary to reset the grid phase as compellingly demonstrated in the authors' analyses of the experimental data.

3) Related to the above point, is the type of learning/plasticity proposed in the model plausible, considering that grid cells are active at most about 50% of the time that a border cell is active? Does the size of the border field matter for the plasticity in the model? It is also unclear how a particular relationship between grid cell and border arises in two dimensions. The authors illustrate their learning algorithm in 1D (Figure 4A). The schematic is very informative but fails to explain clearly how grid cells become associated with a border cell. For cells belonging to the largest grid module (#5), and for which the effect of environmental deformation is the strongest, at most one field is present along a particular wall. This field overlaps only partially with the field of border cells firing for this particular wall. It begs the question of whether the co-firing between the border and the grid cells is actually above 'chance level' (i.e. baseline co-firing) to potentiate the border-to-grid synapse. The size of the border field may thus be determinant for this learning to take place, as suggested in a manuscript published recently (Santos-Pata et al., 2017; not cited). Alternatively, does the relationship between border and grid cells have to be learned? There is a large body of evidence from behavioral and experimental studies that environmental borders reset the path integrator. Can't this effect just be explained by spatially oscillatory neurons (with fixed spatial period) that are reset as soon as animals encounter a border? It might explain the experimental data reanalyzed by the authors without the need of a plasticity process between border and grid cells.

4) Along the same lines as the previous comment, grid cells have been reported to show less gridness along the walls than at the center of the environment. This could result from enhanced drive by border neurons along the boundaries. Given that grid cells in this manuscript are under direct excitatory influence of border cells, why don't they show more firing along the walls?

5) There is a potentially major methodological issue with the quantification of the grid shift in Figure 6A. Even in a familiar environment, the authors find a grid shift of about 7 cm. The question remains of whether this is just an artifact of the running statistics. Grid fields close to one border will be mostly visited if the animal was just contacting this border, whereas the occupancy in the maps conditioned to the other border will be very low. Thus the estimate of these fields will be good in the maps conditioned to the close border and bad in the maps conditioned to the distal border. A proper analysis would require some quantification of how much of a grid shift is predicted by the undersampling of the occupancy at the distal border. Similarly, for the rescaling factors in Figure 6B, the rescaling for the boundary-conditioned case could be influenced by noisy estimates of undersampled firing fields.

---

## [Author Response]

Essential revisions:1) The model is only weakly connected to the data analysis part. As the authors themselves admit in the Discussion, the specific design of the model is not very important (Discussion, second paragraph). The main theoretical idea that border/boundary information resets some path integrating mechanism is actually quite old (and was already concluded from the Gothard et al., 1996 data). This questions whether the present model has any added value, particularly since all model components are well-known in the literature and many of the model assumptions are highly debated (which is not even discussed in the paper). The model should be more clearly presented in the foundation of previous work. Also, the authors should present the results of their data analysis first, and then present the model as a potential mechanistic explanation of the results. In general, they should devote more of the main text to the analysis component (about 3/5 of the supplementary materials are devoted to this part).

As suggested by the reviewers, we have now reorganized the paper such that the reanalyses are presented first, and with more of the main text and figures devoted to these analyses. We have also moved the place cell modeling portion of the paper to the supplement to foreground the grid cell reanalyses and modeling. Finally, we have grounded our work more fully in the literature.

2) The border fields in the model are smaller than typical, firing along only 50% of the border. Although this is not abnormally small, most border cells published in the literature appear to fire along longer segments of a boundary, often times along the entire boundary and extending across a corner. One wonders if the model still works if border cells are incorporated that simulate the range of lengths reported in the literature, with the majority firing along the entire border (or at least >= 75% of the border). Although the analysis of the data based on the authors' insights from the model is the real contribution of this paper, I think it is important for the authors to show whether the model still works with appropriately sized border cells (if for no other reason than to forestall distraction from the experimental analyses if readers are skeptical about the model). If the model fails to produce grid resets with larger border cell fields (perhaps due to ambiguities between the orientation of the grid fields and the extended borders), this would be an important point to discuss, suggesting that other inputs than border cells are necessary to reset the grid phase as compellingly demonstrated in the authors' analyses of the experimental data.

In our previous submission, border fields did extend around corners uniformly tiling the perimeter of the environment; we now show examples of border units with these properties for clarity (Figure 4A). Following the reviewers’ suggestion, we now also include simulations from a rate-based model in which border fields cover the entire length of a wall of the environment (Figure 5—figure supplement 1). These fields were also distributed evenly around the environment perimeter, with the fields of some border units wrapped around corners. This model reproduces boundary-tethered phase shifts, as well as scale-dependent and local distortions of the time-averaged grid pattern, as reported in the main text and our initial submission.

3) Related to the above point, is the type of learning/plasticity proposed in the model plausible, considering that grid cells are active at most about 50% of the time that a border cell is active? Does the size of the border field matter for the plasticity in the model? It is also unclear how a particular relationship between grid cell and border arises in two dimensions. The authors illustrate their learning algorithm in 1D (Figure 4A). The schematic is very informative but fails to explain clearly how grid cells become associated with a border cell. For cells belonging to the largest grid module (#5), and for which the effect of environmental deformation is the strongest, at most one field is present along a particular wall. This field overlaps only partially with the field of border cells firing for this particular wall. It begs the question of whether the co-firing between the border and the grid cells is actually above 'chance level' (i.e. baseline co-firing) to potentiate the border-to-grid synapse. The size of the border field may thus be determinant for this learning to take place, as suggested in a manuscript published recently (Santos-Pata et al., 2017; not cited). Alternatively, does the relationship between border and grid cells have to be learned? There is a large body of evidence from behavioral and experimental studies that environmental borders reset the path integrator. Can't this effect just be explained by spatially oscillatory neurons (with fixed spatial period) that are reset as soon as animals encounter a border? It might explain the experimental data reanalyzed by the authors without the need of a plasticity process between border and grid cells.

We implemented plasticity in our model to derive a consistent mapping of border input to grid phase in the familiar environment. We selected this particular implementation of plasticity because it had been used previously to model connections between grid cells and place cells (Pilly and Grossberg, 2013). With this form of plasticity it is competition among inputs which matters most – thus the size of the border fields as well as the scale and phase of the grid pattern will have some effect on plasticity. In our model, the connection strengths were chosen such that, despite these differences, border inputs were sufficient to induce boundary-tethered shifts in grid phase in all modules during deformations. Other forms of plasticity could likely also work. We also agree with the reviewers that the results presented here do not necessarily depend on plasticity. To demonstrate this, we now include simulations which reproduce grid scale-dependent rescaling without plasticity as a result of boundary-tethered shifts in phase imposed by fiat in the model (Figure 5—figure supplement 2). Nevertheless, there are experimental results beyond the modeled phenomena which suggest that phase-boundary anchoring may be plastic. For example, as an environment becomes more familiar grid scale decreases (Barry et al., 2012), which would lead to conflicting boundary-phase relationships in the absence of plasticity. Likewise, grid phase relative to boundaries can differ across geometrically identical environments (Marozzi, et al., 2015). Finally, rescaling during deformations does not persist indefinitely, as the grid representation relaxes to its familiar scale over time (Barry, et al., 2007). Thus, even if plasticity is ultimately not necessary to capture the phenomena we model here, there is a motivation for hypothesizing that boundary-phase relationships can change over time.

4) Along the same lines as the previous comment, grid cells have been reported to show less gridness along the walls than at the center of the environment. This could result from enhanced drive by border neurons along the boundaries. Given that grid cells in this manuscript are under direct excitatory influence of border cells, why don't they show more firing along the walls?

As the reviewers note, experimentally-observed reduction in gridness along the walls of the environment could be a result of additional border input. This result can be replicated in our model if the connections from border units to grid units are strengthened – when strengthened, the noisy spiking of individual border units is amplified and can be sufficient to override path integration and shift the grid phase parallel to the boundary in undeformed environments. This results in time-averaged grid patterns with increased out-of-field spiking along borders, as observed in two example units from the same module in a simulation with increased border-to-grid connection strength (Author response image 1).

In the model as described in the paper, this disruption of gridness does not occur because we set the strength of border unit input to be: 1) weak enough such that individual noisy border unit inputs are not sufficient on their own to override path integration and induce parallel-to-boundary drift, but also 2) strong enough such that the input from the border unit population as a whole is sufficient to override path integration and shift grid phase. In our view this detail is beyond the scope of the large deformation phenomena we address here, and is for future studies to address. We suggest that future modeling might include increased border-to-grid connection strength in concert with additional inputs such as those derived from visual cues, place cells, or boundary vector cells to capture this grid distortion.

5) There is a potentially major methodological issue with the quantification of the grid shift in Figure 6A. Even in a familiar environment, the authors find a grid shift of about 7 cm. The question remains of whether this is just an artifact of the running statistics. Grid fields close to one border will be mostly visited if the animal was just contacting this border, whereas the occupancy in the maps conditioned to the other border will be very low. Thus the estimate of these fields will be good in the maps conditioned to the close border and bad in the maps conditioned to the distal border. A proper analysis would require some quantification of how much of a grid shift is predicted by the undersampling of the occupancy at the distal border. Similarly, for the rescaling factors in Figure 6B, the rescaling for the boundary-conditioned case could be influenced by noisy estimates of undersampled firing fields.

We thank the reviewers for noting this potentially confounding bias. In the previous draft our shift and rescaling analyses relied only on pixels that were sampled after contact with both opposing boundaries in the hopes of minimizing sampling biases; however, this may be inadequate. We have now changed our measure of shift in two key ways to address this possibility. Firstly, when generating opposing boundary rate maps for the purpose of computing shift we now match the number of samples within each pixel (prior to smoothing) by selecting a random subset of timepoints when the animal was at each location after contacting each boundary. Secondly, because the maximum shift that could possibly be observed is constrained to half the grid scale, we now compute the cross-correlation across the lags of [-grid scale/2 to +grid scale/2], and express shift as a shift-to-scale ratio. Because there is noise introduced by our random selection of timepoints, we repeat this shift calculation for each cell 100 times, and take the average across those repetitions as our final estimate of shift. The shift we observe with this more controlled analysis in which sampling biases are eliminated is qualitatively similar but more reliable (less variance across cells and trials). When computing rescaling, we now also take a similar approach – when comparing rescaling in the wholetrial versus boundary-conditioned data, we subsampled the whole-trial rate map to match the sampling distribution of the to-be-compared boundary rate map, with the mean of 100 repetitions for each comparison taken as the sampling-equated rescaling estimate. As in our original analyses, we find that boundary-conditioning reduces the appearance of rescaling when these sampling biases are eliminated.